Epitope-based chimeric peptide vaccine design against S, M and E proteins of SARS-CoV-2, the etiologic agent of COVID-19 pandemic: an in silico approach

Rahman M. Shaminur 1
Hoque M. Nazmul 1 2
Islam M. Rafiul 1
Akter Salma 1 3
Rubayet Ul Alam A. S. M. 4
Siddique Mohammad Anwar 1
Saha Otun 1
Rahaman Md. Mizanur razu002@du.ac.bd 1
Sultana Munawar 1
Crandall Keith A. 5
Hossain M. Anwar hossaina@du.ac.bd 1 6
1 Department of Microbiology, University of Dhaka , Dhaka , Bangladesh
2 Department of Gynecology, Obstetrics and Reproductive Health, Bangabandhu Sheikh Mujibur Rahman Agricultural University , Gazipur , Bangladesh
3 Department of Microbiology, Jahangirnagar University , Savar , Bangladesh
4 Department of Microbiology, Jashore University of Science and Technology , Jashore , Bangladesh
5 Computational Biology Institute, Milken Institute School of Public Health, George Washington University , Washington , Washington D.C. , United States of America
6 Vice–Chancellor, Jashore University of Science and Technology , Jashore , Bangladesh
Azevedo Vasco
Electronic publication date: 2020 Jul 27
Publication date: 2020
Volume: 8
Electronic Location ID: e9572
Received 2020 Apr 16; Accepted 2020 Jun 29
Copyright: ©2020 Rahman et al.
Copyright year: 2020
Copyright holder: Rahman et al.
License: This is an open access article distributed under the terms of the Creative Commons Attribution License, which permits unrestricted use, distribution, reproduction and adaptation in any medium and for any purpose provided that it is properly attributed. For attribution, the original author(s), title, publication source (PeerJ) and either DOI or URL of the article must be cited.
License URL: https://creativecommons.org/licenses/by/4.0/

Keywords: SARS-CoV-2, Muti-epitope, Chimeric Peptide Vaccine, B-cell Epitope, T-cell Epitope

Funding: The authors received no funding for this work.

==============================
Severe acute respiratory syndrome coronavirus 2 (SARS-CoV-2) is the etiologic agent of the ongoing pandemic of coronavirus disease 2019 (COVID-19), a public health emergency of international concerns declared by the World Health Organization (WHO). An immuno-informatics approach along with comparative genomics was applied to design a multi-epitope-based peptide vaccine against SARS-CoV-2 combining the antigenic epitopes of the S, M, and E proteins. The tertiary structure was predicted, refined and validated using advanced bioinformatics tools. The candidate vaccine showed an average of ≥90.0% world population coverage for different ethnic groups. Molecular docking and dynamics simulation of the chimeric vaccine with the immune receptors (TLR3 and TLR4) predicted efficient binding. Immune simulation predicted significant primary immune response with increased IgM and secondary immune response with high levels of both IgG1 and IgG2. It also increased the proliferation of T-helper cells and cytotoxic T-cells along with the increased IFN-γ and IL-2 cytokines. The codon optimization and mRNA secondary structure prediction revealed that the chimera is suitable for high-level expression and cloning. Overall, the constructed recombinant chimeric vaccine candidate demonstrated significant potential and can be considered for clinical validation to fight against this global threat, COVID-19.

Introduction

Emergence of the SARS-CoV-2, which was first reported in Hubei Province of Wuhan, China in December 2019, is responsible for the ongoing global pandemic of coronavirus disease 2019 (COVID-19) spreading across 216 countries, areas, or territories with 6,992,274 active infection cases and 403,128 deaths until June 9, 2020 (WHO, 2020). This SARS-CoV-2 is the third coronovairus (CoV) belonging to Genus Betacoronavirus that can infect human after the two previously reported coronavirus- severe acute respiratory syndrome (SARS-CoV) (Almofti et al., 2018; Wu et al., 2020), and Middle East respiratory syndrome (MERS-CoV) (Badawi et al., 2016; Pallesen et al., 2017; Ul Qamar et al., 2019). The SARS-CoV-2 has a positive-sense, single-stranded, and ∼30 kilobase long RNA genome showing 79.0% and 50.0% identity to the genomes of SARS-CoV and MERS-CoV, respectively (Abdelmageed et al., 2020; Tang et al., 2020; Lu et al., 2020; Hoque et al., 2020a). Among multiple encoded proteins (structural, non-structural, and accessory), four major structural proteins are the spike (S) glycoprotein, small envelope protein (E), membrane protein (M), and nucleocapsid protein (N) (Ahmed, Quadeer & McKay, 2020).

The S glycoprotein, because of its higher antigenicity and surface exposure (Almofti et al., 2018; Zhou et al., 2019; Shang et al., 2020), plays the most crucial role for the attachment and entry of viral particles into the host cells through the host angiotensin-converting enzyme 2 (ACE2) receptor (Gralinski & Menachery, 2020; Shang et al., 2020). It is noteworthy that E and M proteins also have important functions in the viral assembly, budding and replication of virus particles, as well as play role in augmenting the immune response against SARS-CoV (Shi et al., 2006; Schoeman & Fielding, 2019; Shang et al., 2020). Monoclonal antibodies (mAbs) primarily target the trimeric S glycoprotein of the virion consisting of three homologous chains (A, B, and C), and this protein is composed of two major domains, the receptor-binding domain (RBD) and the N-terminal domain (NTD) (Pallesen et al., 2017; Song et al., 2018; Zhou et al., 2019; Wrapp et al., 2020). The NTD is located on the side of the spike trimer and has not been observed to undergo any dynamic conformational changes (Shang et al., 2020). Thus, this specific region might play role in viral attachment, inducing neutralizing antibody responses and stimulating a protective cellular immunity (Almofti et al., 2018; Ul Qamar et al., 2019; Shang et al., 2020).

Most of the recent vaccine candidates induce neutralizing antibodies against the different forms and/or variants of the spike protein of the SARS-CoV-2 (Le et al., 2020). However, the immune responses generated from using single protein have generally been inadequate to warrant their use in the development of an effective prophylactic tool (Shi et al., 2015; Shey et al., 2019). On this note, multi-epitope vaccine candidates have already been designed against several viruses, including MERS-CoV and SARS-CoV, and their efficacies have been further reported (Almofti et al., 2018; Ul Qamar et al., 2019; Yong et al., 2019). Two related studies have reported the in-silico design of epitope based chimeric vaccine candidates targeting E, M, S and N proteins of SARS-CoV-2, albeit not peer-reviewed (Yazdani et al., 2020; Akhand et al., 2020). Besides, Kibria, Ullah & Miah (2020) performed an immunoinformatic approach to design a 70 aa long multi-epitope vaccine focusing on the the virion outer surface proteins (E, M, and S) (Chan et al., 2020).

Scientists are racing over the clock to develop effective vaccine for controlling and preventing COVID-19 based on the genomics, functional structures, and host-pathogen interactions; nevertheless, the ultimate results of these efforts is yet uncertain. Currently, 10 candidate vaccines against SARS-CoV-2 are in the clinical trial, and 121 more under preclinical evaluation (WHO, 2020). Strikingly, researchers are trialing different technologies, albeit targeting spike protein mainly, some of which have not been used in a licensed vaccine before (Le et al., 2020). Their vaccine appears to be effective and safe based on a limited data and application of the vaccine within a relatively tiny group of individuals. However, there are several uncertainties, for example, whether it will relate to antibody responses in the general population, be safe within a specific sub-population (children, pregnant women, and elder people), as well as lack of a standardized virus neutralization assay and accurate vaccine titer are complicating data interpretation. Moreover, only half of the medium dose receiver developed neutralizing antibody and the T-cell response is not particularly impressive (Sheridan, 2020).

Thus, mimicking of a more natural state of the virus where surface exposed proteins, or the immunodominant epitopes of those proteins influencing the immune response might be a solution, if those candidates ultimately cannot meet the final goal. Furthermore, excluding the nucleocapsid (N) protein, which is embedded within the structure and attached to the viral genome, will amplify the chance of developing a more pseudo-virus state for the expressed chimera that may produce specific antibody as well as T-cell responses. Furthermore, peptide-based chimeric vaccines are biologically safe as they do not need in vitro virus culture, and their selectivity might ensure accurate activation of specific immune responses (Dudek et al., 2010; Wang et al., 2019).

Considering the facts, we have proposed the development of a multi-epitope vaccine candidate, which differs from all the previous studies in the aspect of containing whole RBD and NTD regions of the spike protein alongwith specific epitopes of M and E proteins, giving an excellent chimeric conformation and might lead to the generation of a more potent protective immune responses since smaller epitopes have less ability to give better immune protection. Hence, we can assume that chimeric vaccine targeting multiple epitopes on the RBD and NTD segments of the S protein, M and E proteins would be a potentially effective vaccine candidate in combatting COVID-19 pandemic, and therefore, could be used against the could be used against the highly pathogenic SARS-CoV-2.

Results

Comparative structural analysis of SARS-CoV, MERS-CoV and SARS-CoV-2

Multiple sequence alignment revealed that the S protein of SARS-CoV-2 shares 77.38% and 31.93% sequence identity with the S proteins of the SARS-CoV and MERS-CoV, respectively (Fig. S1). The structural (validated using the Ramachandran plot as in Fig. S2) alignment of the coronavirus S proteins reflects high degree of structural heterogeneity in the receptor-binding domain (RBD) and N-terminal domain (NTD) of the chain A and chain C compared to that of chain B (Fig. S3). Divergence of individual structural domains, NTD and RBD of 2019-nCoV spike protein from both of the SARS-CoV and MERS-CoV warrants the domains for epitope-based chimeric vaccine development, particularly against SARS-CoV-2.

Screening for B-cell epitopes

Linear epitopes prediction (ElliPro) based on solvent-accessibility and flexibility revealed 15, 18, and 19 epitopes within the chain A, B and C of S protein, respectively wherein score >0.8 was the threshold for the highly antigenic epitopes (Table 1). The amino acid (aa) residues in 56-194 and 395-514 position of the detected epitopes belonged to RBD and NTD regions of the S protein, respectively. However, the epitopes with aa position of 1067-1146 were not selected as the potential epitope candidate because of their presence in viral transmembrane domains (Fig. S4). The tertiary structures of the RBD and NTD illustrate their surface-exposed position on the S protein (Fig. 1). Using IEDB analysis resource and Bepipred linear epitope prediction 2.0 tools, we predicted eight and six B-cell epitopes in RBD and NTD regions out of total 22 epitopes in S protein, while the E and M proteins had 2 and 6 epitopes, respectively (Fig. 2, Table S1). However, only 5 epitopes were exposed on the surface of the virion, and had a high antigenicity score (>0.4), indicating their potentials in initiating immune responses (Table 2).

Table 1 Linear epitopes present on spike (S) glycoprotein surface predicted through ElliPro in IEDB-analysis resource based upon solvent-accessibility and flexibility are shown with their antigenicity scores.

The highlighted green coloured regions were the potential antigenic domains while the yellow coloured region represents the trans-membrane domain of the S protein.

No.	Chain	Start	End	Peptide	Residues	Score	
1	A	395	514	VYADSFVIRGDEVRQIAPGQTGKIADYNYKLP DDFTGCVIAWNSNNLDSKVGGNYNYLYRLFRK SNLKPFERDISTEIYQAGSTPCNGVEGFNCYFPLQSYG FQPTNGVGYQPYRVVVLS	120	0.837	
2	58	194	FFSNVTWFHAIHVSGTNGTKRFDNPVLPFNDGVYFAS TEKSNIIRGWIFGTTLDSKTQSLLIVNNATNVVIKVCEF QFCNDPFLGVYYHKNNKSWMESEFRVYSSANNCTFEYV SQPFLMDLEGKQGNFKNLREFVF	137	0.835	
3	1067	1146	YVPAQEKNFTTAPAICHDGKAHFPREGVFVSNG THWFVTQRNFYEPQIITTDNTF VSGNCDVVIGIVNNTVYDPLQPELD	80	0.83	
4	201	270	FKIYSKHTPINLVRDLPQGFSALEPLVDLPIG INITRFQTLLALHRSYLTPGDSSSGWTAGAAAYYVGYL	70	0.76	
5	331	381	NITNLCPFGEVFNATRFASVYAWNRKRISN CVADYSVLYNSASFSTFKCYG	51	0.706	
6	700	720	GAENSVAYSNNSIAIPTNFTI	21	0.668	
7	27	35	AYTNSFTRG	9	0.66	
8	909	936	IGVTQNVLYENQKLIANQFNSAIGKIQD	28	0.633	
9	789	813	YKTPPIKDFGGFNFSQILPDPSKPS	25	0.6	
10	623	642	AIHADQLTPTWRVYSTGSNV	20	0.598	
11	891	907	GAALQIPFAMQMAYRFN	17	0.591	
12	579	583	PQTLE	5	0.551	
13	687	692	VASQSI	6	0.55	
14	653	659	AEHVNNS	7	0.539	
15	679	684	NSPRRA	6	0.521	
16	B	1067	1146	YVPAQEKNFTTAPAICHDGKAHFPREGVFVSNGTH WFVTQRNFYEPQIITTDNTFVS GNCDVVIGIVNNTVYDPLQPELD	80	0.826	
17	89	194	GVYFASTEKSNIIRGWIFGTTLDSKTQSLLIVNNATN VVIKVCEFQFCNDPFLGVYYHKNNKSWMESEFRVYS SANNCTFEYVSQPFLMDLEGKQGNFKNLREFVF	106	0.816	
18	58	87	FFSNVTWFHAIHVSGTNGTKRFDNPVLPFN	30	0.81	
19	203	270	IYSKHTPINLVRDLPQGFSALEPLVDLPIGINIT RFQTLLALHRSYLTPGDSSSGWTAGAAAYYVGYL	68	0.748	
20	465	509	ERDISTEIYQAGSTPCNGVEGF NCYFPLQSYGFQPTNGVGYQPYR	45	0.727	
21	436	458	WNSNNLDSKVGGNYNYLYRLFRK	23	0.672	
22	700	720	GAENSVAYSNNSIAIPTNFTI	21	0.671	
23	27	35	AYTNSFTRG	9	0.666	
24	909	9036	IGVTQNVLYENQKLIANQFNSAIGKIQD	28	0.641	
25	624	643	IHADQLTPTWRVYSTGSNVF	20	0.617	
26	328	365	RFPNITNLCPFGEVFNATRFASVYAWNRKRISNCVADY	38	0.608	
27	891	907	GAALQIPFAMQMAYRFN	17	0.602	
28	577	583	RDPQTLE	7	0.598	
29	790	817	KTPPIKDFGGFNFSQILPDPSKPSKRSF	28	0.595	
30	673	693	SYQTQTNSPRRARSVASQSII	21	0.567	
31	526	537	GPKKSTNLVKNK	12	0.553	
32	653	661	AEHVNNSYE	9	0.548	
33	554	563	ESNKKFLPFQ	10	0.52	
34	C	56	194	LPFFSNVTWFHAIHVSGTNGTKRFDNPVLPFNDGVYFA STEKSNIIRGWIFGTTLDSKTQSLLIVNNATNVVIKV CEFQFCNDPFLGVYYHKNNKSWMESEFRVYSSANNCTFE YVSQPFLMDLEGKQGNFKNLREFVF	139	0.84	
35	1067	1146	YVPAQEKNFTTAPAICHDGKAHFPREGVFV SNGTHWFVTQRNFYEPQIITTDNTFVSGN CDVVIGIVNNTVYDPLQPELD	80	0.822	
36	201	270	FKIYSKHTPINLVRDLPQGFSA LEPLVDLPIGINITRFQTLLALHR SYLTPGDSSSGWTAGAAAYYVGYL	70	0.77	
37	27	35	AYTNSFTRG	9	0.676	
38	465	509	ERDISTEIYQAGSTPCNGVEGFNCY FPLQSYGFQPTNGVGYQPYR	45	0.675	
39	700	720	GAENSVAYSNNSIAIPTNFTI	21	0.658	
40	909	936	IGVTQNVLYENQKLIANQFNSAIGKIQD	28	0.633	
41	437	458	NSNNLDSKVGGNYNYLYRLFRK	22	0.629	
42	673	684	SYQTQTNSPRRA	12	0.619	
43	790	817	KTPPIKDFGGFNFSQILPDPSKPSKRSF	28	0.602	
44	891	907	GAALQIPFAMQMAYRFN	17	0.597	
45	578	583	DPQTLE	6	0.589	
46	620	631	VPVAIHADQLTP	12	0.579	
47	329	362	FPNITNLCPFGEVFNATRFASVYAWNRKRISNCV	34	0.571	
48	687	692	VASQSI	6	0.566	
49	835	845	KQYGDCLGDIA	11	0.567	
50	653	659	AEHVNNS	7	0.559	
51	527	536	PKKSTNLVKN	10	0.546	
52	635	642	VYSTGSNV	8	0.51	

Among the five annotated epitopes having antigenicity score of ≥ 0.5 (VaxiJen 2.0 tool), RBD and NTD regions each possessed two highly antigenic epitopes while the envelope (E) protein contained only one highly antigenic epitope and membrane (M) protein has none (Table 2). Furthermore, the Kolaskar and Tongaonkar antigenicity profiling found five highly antigenic epitopes in RBD region with an average (antigenicity) score of 1.042 (minimum = 0.907, maximum = 1.214), and seven highly antigenic epitopes in NTD with an average (antigenicity) score of 1.023 (minimum = 0.866, maximum = 1.213) (Fig. S5, Table S2). The average Kolaskar scores for envelope protein B-cell epitope (EBE) and membrane protein B-cell epitope (MBE) were 0.980 and 1.032, respectively (Table S2). However, through ABCPred analysis, we further verified 18 and 11 B-cell epitopes in RBD and NTD regions with average antigenicity score of 0.775 and 0.773 in the associated domains, respectively (Table S3).

Selection of T-cell and IFN-γ inducing epitopes

The IEDB MHC-I prediction tool retrieved 77 T-cell epitopes in RBD that interacted with 21 possible MHC-I alleles whereas the NTD domain possessed 35 T-cell epitopes with 17 possible MHC-I alleles (Data S1). Similarly, the IEDB MHC-II prediction tool generated 13-mer 124 peptides from the RBD, and 10-mer 73 peptides in the NTD segments of the S protein that showed interaction with many different and/or common MHC-II alleles with an IC50 value ranging from 1.4 to 49.9 nM (Data S1). Furthermore, the analysis tool of the IEDB generated an overall scores for proteasomal processing, TAP transport, and MHC-binding efficiency indicating the intrinsic potential of the epitopes to be recognized by immunoreactive T-cells (Data S1).

Figure 1 The three-dimensional (3D) structure of the N-terminal domains (NTDs) and receptor binding domains (RBDs) of the spike (S) proteins of SARS-CoV-2 (surface view).

The orange, cyan, and yellow colored regions represent the potential antigenic domains predicted by the IEDB analysis resource ElliPro analysis.

Figure 2 Predicted B-cell epitopes using BepiPred-2.0 epitope predictor in IEDB-analysis resource web-based repository.

Yellow areas above threshold (red line) are proposed to be a part of B cell epitopes in (a) RBD and (b) NTD regions of S protein, (c) envelop (E) and (d) membrane (M) proteins of SARS-CoV-2.

Table 2 B-cell epitopes predicted using Bepipred linear epitope prediction 2.0 in IEDB analysis resource web-server along with their start and end positions, average score, and VaxiJen 2.0 determined antigenicity scores.

Domain/ proteins	Position	Sequences	Average Score	Antigenicity	
RBD	341-342	VF	0.502	–	
344-349	ATRFAS	0.520	−0.151	
351-363	YAWNRKRISNCVA	0.522	0.394	
372-378	ASFSTFK	0.527	0.087	
382	V	0.464	–	
402-427	IRGDEVRQIAPGQTGKIADYNYKLPD	0.575	0.932	
440-485	NLDSKVGGNYNYLYRLFRKSN LKPFERDISTEIYQAGSTPCNGVEG	0.554	0.210	
493-516	QSYGFQPTNGVGYQ	0.535	0.670	
NTD	72-81	GTNGTKRFDN	0.573	0.667	
110-113	LDSK	0.511	–	
146-155	HKNNKSWMES	0.573	0.174	
161-162	SS	0.503	–	
164	N	0.499	–	
172-191	SQPFLMDLEGKQGNFKNLRE	0.553	0.749	
MBE	199-218	YRIGNYKLNTDHSSSSDNIA	0.614	0.222	
EBE	57-71	YVYSRVKNLNSSRVP	0.565	0.449	

The findings of IFNepitope program suggests that, both the target RBD and NTD regions of S protein, and membrane protein B-cell linear epitope (MBE) had great probability to release of IFN-γ with a positive score. A total of 56 potential positive IFN-γ inducing epitopes (15-mer) were predicted for the RBD domain with an average epitope prediction score of 0.255 and the maximum SVM score of 0.625. On the other hand, a total of 33 potential positive epitopes were predicted for the NTD domain with an average epitope prediction score of 0.312 and the maximum SVM score of 0.811. Moreover, the M protein also possessed several IFN-γ inducing epitopes having an average epitope prediction score of 0.980 (Table S4).

Design-construction, antigenicity and physicochemical properties of the chimeric vaccine candidate

The selected epitope-sequences for designing of chimeric construct were PADRE (13 aa), MBE (20 aa), NTD (139 aa), RBD (200 aa), EBE (15 aa), and Invasin (16 aa), and the construct was named as CoV-RMEN (417 aa) as shown in Fig. 3A. These segments were connected with a repeat of hydrophobic (glycine; G) and acidic aa (glutamic acid; E) linkers for making the final vaccine construct more flexibile with balanced ratio of acidic and basic amino acids. The molecular weight (MW) of the CoV-RMEN was 46.8 kDa with a predicted isoelectric point (pI) of 8.71. The projected half-life was 4.4 h in mammalian reticulocytes in vitro, and >20 h in yeast and >10 hours in E. coli in vivo. The protein was predicted to be less soluble upon expression with a solubility score of 0.330. An instability index (II) value 29.74 predicted the protein as stable (II of >40 indicates instability). The estimated aliphatic index was 66.59, indicating thermostability of the final chimera. The predicted grand average of hydropathicity (GRAVY) was −0.300. The antigenicity score of 0.450 was predicted by the VaxiJen 2.0 server with a virus model at a threshold of 0.4, and further verified by ANTIGENpro showing score of 0.875 (maximum expected score ranking is 1.0) indicating the high antigenic nature of the designed vaccine, CoV-RMEN. Moreover, the vaccine was also predicted to be non-allergenic on both the AllerTOP v.2 and AllergenFP servers.

Figure 3 Design, construction and structural validation of multi-epitope vaccine candidate (CoV-RMEN) for SARS-CoV-2.

(A) Structural domains and epitopes rearrangement of CoV-RMEN, (B) secondary structure of CoV-RMEN as analyzed through CFSSP: Chou and Fasman secondary structure prediction server , (C) final tertiary structure of CoV-RMEN (surface view) obtained from homology modelling on Phyre2 in which domains and epitopes are represented in different colors (PADRE-smudge; membrane B-cell epitope, MBE-magenta; N-terminal domain, NTD-orange; receptor-binding domain, RBD-cyan; envelop B-cell epitope, EBE-blue; invasin-yellow), (D) validation of the refined model with Ramachandran plot analysis showing 94.7%, 4.8% and 0.5% of protein residues in favored, allowed, and disallowed (outlier) regions respectively, (e) ProSA-web, giving a Z-score of −6.17, and (f) the finally predicted primary structure of the CoV-RMEN.

Structural characterization of the CoV-RMEN

The CoV-RMEN peptide was predicted to contain 43.2% alpha helix, 67.4% beta sheet, and 12% turns (Fig. 3B, Fig. S6) using CFSSP:Chou and Fasman secondary structure prediction server. Additionally, regarding the solvent accessibility of aa residues, 34%, 30% and 34% were predicted to be exposed, medium exposed and buried respectively (Fig. S7). The RaptorX Property server predicted only two aa residues in the disordered domains. The Phyre2 server predicted the tertiary structure model of the designed chimeric protein in 5 templates (c5x5bB, c2mm4A, c6vsbB, c5x29B and c6vybB) based opon heuristics to maximize confidence, alignment coverage and percent identity. The final 3D structure of the CoV-RMEN peptide modelled at 82% with more than 90% confidence (Fig. 3C). Moreover, 65 residues were modelled by ab initio. The selected structural model has parameters of RMSD (0.414), GDT-HA (0.9538), and MolProbity (2.035). The Ramachandran plot analysis of the finally modelled protein exhibited 94.7% of the aa residues in favored regions (Fig. 3D), consistent with the 94.0% score predicted by the GalaxyRefine analysis. Additionally, 4.8% of the residues located in allowed regions, and only 0.5% in disallowed regions (Fig. 3D). The chosen model after refinement had an overall quality factor of 74.45% with ERRAT (Fig. S8) and a ProSA-web based Z-score of −6.17 (Fig. 3E).

Molecular docking and dynamics simulation analysis

Among the selected epitopes from the RBD and NTD segments, top five based on IC50 score ( Data S1) revealed highly favorable molecular interaction for stable binding with their respective HLA alleles. Docking complexes thus formed have significantly negative binding affinity (ΔG always remained ≤−8.2 kcal mol−1, average = −9.94 kcal mol−1), and most of the amino acid (aa) residues of the epitopes were involved in molecular interactions with their respective HLA alleles (Fig. 4, Data S1). The immune responses of TLR2, TLR3 and TLR4 against vaccine construct (CoV-RMEN) were estimated by analyzing the overall conformational stability of vaccine protein-TLRs docked complexes. The active interface aa residues of refined complexes of CoV-RMEN and TLRs were predicted (Fig. 5, Table 3).

The relative binding free energies (ΔG) of the protein-TLRs complexes were significantly negative (Table 3) which suggest that the interaction of the chimeric protein might favor stimulation of the TLR receptors. Consistently, the number of contacts made at the interface (IC) per property (ICs charged-charged: 5, ICs charged-polar: 2, ICs charged-apolar: 17, polar-polar: 1, ICs polar-apolar: 7 and apolar-apolar: 16) for the vaccine protein-TLR2 complex. Interface contacts (IC) per property (ICs charged-charged: 16, ICs charged-polar: 22, ICs charged-apolar: 26, polar-polar: 6, ICs polar-apolar: 25 and apolar-apolar: 29) were for the vaccine protein-TLR3 complex. Also, vaccine protein-TLR4 complex showed similar (ICs) per property (ICs charged-charged: 5, ICs charged-polar: 11, ICs charged-apolar: 30, polar-polar: 4, ICs polar-apolar: 31 and apolar-apolar: 39). Furthermore, the molecular dynamics (MD) simulation analysis of the docked CoV-RMEN-TLR3 and CoV-RMEN-TLR4 complexes showed soundly stable RMSD values between ∼4.35 and ∼5.4 nm for a specified time frame of 100 ps at the reasonably consistent temperature (∼300 K) and pressure (1bar), whereas CoV-RMEN-TLR2 complex showed RMSD value between 5.5 and 6.2 with same cut-off parameters. These data validated that the docked complexes (CoV-RMEN-TLR3 and CoV-RMEN -TLR4) are more stable than CoV-RMEN-TLR2 (Fig. 5).

Immune simulation

The cumulative results of immune responses after three times antigen exposure with four-week interval each time revealed that the primary immune response against the antigenic fragments was elevated indicated by gradual increase of IgM level after each antigen exposure (Fig. 6A). Besides, the secondary immune response, crucial for immune stability, have been shown as increased with adequate generation of both IgG1 and IgG2. Also, the elevated level of all circulating immunoglobulins indicates the accuracy of relevant clonal proliferation of B-cell and T-cell population. The level of cytokines after antigen exposure increased concomitantly reflected by escalation of IFN-γ and IL-2, which are most significant cytokines for anti-viral immune response and clonal selection (Fig. 6B). The abundance of different types of B-cells and T-cells, like antigen processing B-cells, resting memory B- and T-cells, B-cells with IgM and IgG remains significantly higher indicating development of immune memory and consequently increased clearance of antigen after exposure (Figs. 6C and 6D). Additionally, T-helper cells and cytotoxic T-cells were found with a drastic up-regulation of Th1 concentration enhancing the B-cell proliferation and immune memory development (Figs. 6E and 6F). The high level of immunoglobulin IgG1 + IgG2, active B-cell and T-helper cell population reflected the development of strong immune response reinforcing the indelible and peerless antigenicity of the CoV-RMEN vaccine candidate.

Figure 4 Molecular docking of top five MHC-I and MHC-II epitopes of RBD and NTD domains with respect to HLA allele binders.

(A–E) and (K–O) represent the top five MHC-I epitopes of RBD and NTD domains, respectively. (F–J) and (P–T) represent the top five MHC-II epitopes binds of the same domains binds to their respective HLA alleles. The protein-peptide docking was performed in GalaxyWEB-GalaxyPepDock-server followed by the refinement using GalaxyRefineComplex and free energy (ΔG) of each complex was determined in PRODIGY server. Ribbon structures represent HLA alleles and stick structures represent the respective epitopes. Light color represents the templates to which the alleles and epitopes structures were built. Further information on molecular docking analysis is also available in Data S1.

Figure 5 Molecular docking and dynamics of CoV-RMEN vaccine with immune receptors (TLR2, TLR3 and TLR4).

Docked complexes for (A) CoV-RMEN and TLR2, (B) CoV-RMEN and TLR3, and (C) CoV-RMEN and TLR4. Magnified interfaces of the respective complexes are figured to (D), (E) and (F) respectively. Active residues of CoV-RMEN colored magenta, and of TLRs colored orange with stick view. ΔG represents the binding affinity of the complexes. Molecular dynamics simulation study of (G) CoV-RMEN and TLR2, (H) CoV-RMEN and TLR3, and (I) CoV-RMEN and TLR4 complexes across the time window of 100 ps. The reasonably invariable RMSD value indicates a stable complex formation.

Table 3 Active interface amino acid residues and binding scores among Toll Like Receptors (TLRs) and the constructed vaccine CoV-RMEN.

Active residues of TLRs	Active residues of CoV-RMEN	HADDOCK score	ΔG (kcal mol−1)	
TLR-2	V536, C537, S538, C539, E540, S543, E547, P567, R569, L570	D72, Y75, L101, I103, I112, C150, F152, E153, Y154, V155, S156, F176, F178, R181, F182, L371	−30.4	−9.0	
TLR-3	D36, H39, K41, R643, F644, P646, F647, T650, C651, E652, S653, I654, W656, F657, V658, N659, W660, I661, N662, E663	F43, S44, N45, V46, T47, W48, D72, Y75, F76, L101, I103, I112, F152, E153, Y154, V155, S156, Q157, F159, F178, R181, F182, L371	−47.2	−14.9	
TLR-4	P53, F54, S55, H68, G70, Y72, S73, F75, S76, Q99, S102, G124	G39, L40, D72, Y75, F76, F90, L101, I103, I112, F152, Y154, S156, Q157, F159, R174, E175, F176, F178, R181, F182, P183, L371, P374, P380, G381	−52.1	−16.0	

Figure 6 C-ImmSim presentation of an in silico immune simulation with the chimeric peptide.

(A) The immunoglobulins and the immunocomplex response to antigen (CoV-RMEN) inoculations (black vertical lines); specific subclasses are indicated as colored peaks, (B) concentration of cytokines and interleukins, and inset plot shows danger signal together with leukocyte growth factor IL-2, (C) B-cell populations after three injections, (D) evolution of B cell, (E) T-helper cell populations per state after injections, and (F) evolution of T-helper cell classes with the course of vaccination.

Population coverage analysis

The selected CTL and HTL epitopes covered 94.9% and 73.11% of the world population, respectively. Importantly, CTL and HTL epitopes showed 98.63% population coverage worldwide when considered in combination. The highest population coverage was found to be 99.99% in the Latin American country, Peru (Fig. 7, Data S2). In China, where the viral strain (SARS-CoV-2) first appeared and had more devastating outbreaks, the population coverage for CTL and HTL epitopes was 92.67% and 53.44%, respectively with a combined coverage of 96.59%. SARS-CoV-2 is currently causing serious pandemics in different continents of the globe including Italy, England, Spain, Iran, South Korea and United States of America where the combined population coverage was found to be 98.8%, 99.44%, 95.35%, 98.48%, 99.19% and 99.35%, respectively (Fig. 7A, Data S2). In addition to geographical distribution, the ethnic groups also found to be an important determinant for good coverage of the CTL and HTL epitopes (Fig. 7B). Of the studied 147 ethnic groups, the Peru Amerindian had highest population coverage for CTL (99.98%) while the HTL epitopes had highest population coverage for Austria Caucasoid (88.44%) (Fig. 7B, Data S2). Furthermore, 53.06% of the ethnic groups had a combined population coverage of more than 90.0% for both CTL and HTL epitopes.

Figure 7 Population coverage of the selected T-cell epitopes and their respective HLA alleles.

The circular plot illustrates the relative abundance of the top 70 geographic regions and ethnic groups for selected CTL and HTL epitopes, which were used to construct the vaccine and their corresponding MHC HLA alleles were obtained for population coverage analysis both individually (either MHC-I or MHC-II) and in combination (MHC-I and MHC-II). (A) Population coverage of top seventy geographical regions out of 123 regions. (B) Population coverage of top seventy ethnic groups selected from 146 ethnic groups. Regions and ethnic groups in the respective MHC-I and MHC-II epitopes are represented by different colored ribbons, and the inner blue bars indicate their respective relative coverages. Further information on population coverage analysis is also available in Data S1.

Expression prediction of the CoV-RMEN

The length of the optimized codon sequence of the vaccine construct CoV-RMEN in E. coli (strain K12) was 1,251 nucleotides. The optimized nucleotide sequence had a Codon Adaptation Index (CAI) of 0.87, and the average GC content of 50.26% showing the possibility of good expression of the vaccine candidate in the E. coli host (Figs. 8A–8C). Moreover, the evaluation of minimum free energy for 25 structures of chimeric mRNA through Mfold’server showed that ΔG of the best predicted structure for the optimized construct was ΔG = −386.50 kcal/mol. The first nucleotides at 5′did not have a long stable hairpin or pseudoknot. Therefore, the binding of ribosomes to the translation initiation site, and the following translation process can be readily accomplished in the target host. These outcomes were in the agreement with data obtained from the ‘RNAfold’ web server (Figs. 8D and 8E) where the free energy was −391.37 kcal/mol.

Figure 8 Codon optimization and mRNA structure of CoV-RMEN gene for expression in E. coli.

(A) GC curve (average GC content: 50.26%) of the optimized CoV-RMEN gene, (B) percentage distribution of codons in computed codon quality groups, (C) relative distribution of codon usage frequency along the gene sequence to be expressed in E. coli, and codon adaptation index (CAI) was found to be 0.87 for the desired gene, (D) secondary structure and stability of corresponding mRNA, and (E) resolved view of the start region in the mRNA structure of CoV-RMEN.

After codon optimization and mRNA secondary structure analysis, the sequence of the recombinant plasmid was designed by inserting the adapted codon sequences into pETite vector (Lucigen, USA), which contains SUMO (Small Ubiquitin-like Modifier) tag and 6x-His tag facilitating both the solubilization and affinity purification of the recombinant protein using SnapGene software (Fig. 9). As alternative to E. coli for the expression system, HEK-293 eukaryotic cell line found promising for CoV-RMEN expression. The codon adaption index (CAI), GC content for this system were 1.0 and 61.60 respectively, which indicate high level of expression of the vaccine construct in the HEK-293 cell line as well.

Figure 9 In silico fusion cloning of the CoV-RMEN.

The final vaccine candidate sequence was inserted into the pETite expression vector where the red part represents the gene coding for the predicted vaccine, and the black circle represents the vector backbone. The six His-tag and SUMU-tag are located at the Carboxy-terminal end.

Discussion

SARS-CoV-2, the virus with high zoonotic importance and transmission rate, has spread rapidly around the world and causes life-threatening COVID-19 (Gorbalenya et al., 2020). The number of SARS-CoV-2 infections, and subsequent deaths are increasing day by day (Tai et al., 2020), and thus, COVID-19 outbreak was declared as a public health emergency by the International Concerns (Hui et al., 2020; Zhou et al., 2020a). Scientific community across the world is trying to develop an effective and safe vaccine against this rapidly emerging SARS-CoV-2 (Abdelmageed et al., 2020). Although a good number of vaccine candidates for COVID-19 are now under trials, some of them are advanced to human trials (Lane, 2020), none has yet been declared to be effective and safe for prevention of SARS-CoV-2 infections. Strikingly, no effective therapeutic drugs or vaccines are yet to be available for the treatment of SARS-CoV-2 patients (Hoque et al., 2020a). Through a comprehensive genomic and proteomic study, we endeavor to design an antigenic multi-epitope (immunodominant) chimeric vaccine for SARS-CoV-2, named as CoV-RMEN (417 aa), which will nullify the involvement of lab-escape viral transmission, reduce the cost, and may elicit immunity by selectively stimulating antigen-specific B- and T-cells.

The novel approach of multi-epitope based (includes conserved multiple epitopes) vaccines designing represents inducing specific cellular immunity, and highly potent neutralizing antibodies against infections (Dawood et al., 2019; Yong et al., 2019; Gralinski & Menachery, 2020; Kibria, Ullah & Miah, 2020). These epitope-based vaccines also provide increased safety and have the ability to focus on sustainable immune responses because of including conserved multiple epitopes. Unlike the full-length S protein, the RBD and NTD segments possess critical neutralizing domains without any non-neutralizing immunodominant region (Ul Qamar et al., 2019; Gralinski & Menachery, 2020; Shang et al., 2020; Wrapp et al., 2020). Mutations on the RBD may enable the new strains to escape neutralization by established RBD-targeting antibodies, hence other functional regions, especially the NTD, should be considered for developing an effective vaccine as well (Wang et al., 2019; Zhou et al., 2019). Besides, combined administration of RBD and NTD proteins induced highly potent neutralizing antibodies and long-term protective immunity in animal models (Song et al., 2018). Considering the safety and effectiveness perspectives, the RBD and NTD are more promising candidates in the development of SARS-CoV-2 vaccines over the full-length S protein. The presence of E and M proteins on the envelope can augment the immune response against SARS-CoV (Millet & Whittaker, 2015; Almofti et al., 2018) and thus, considered for suitable candidate for vaccine development (Yong et al., 2020; Ahmed, Quadeer & McKay, 2020; Gralinski & Menachery, 2020). Thus, antibodies against the immunologically substantial epitopes of S, M and E proteins of SARS-CoV-2 would provide protective immunity to the infection (Yong et al., 2020; Ahmed, Quadeer & McKay, 2020; Gralinski & Menachery, 2020; Shang et al., 2020). Therefore, the immune response targeting the RBD and/or NTD of the S glycoprotein, M and E proteins of SARS-CoV-2 would be an important prophylactic and therapeutic interventions, which can be tested further in suitable models before clinical trials (Chan et al., 2020).

Effective immunity to viral infections is significantly dependent on activation of both B- and T-cells (Shi et al., 2015; Shey et al., 2019). Therefore, inducing specific humoral or cellular immunity against pathogens, an ideal vaccine should contain both B-cell and T-cell epitopes. Our analyses revealed that selected RBD and NTD regions of the CoV-RMEN contain ample amount of high-affinity B-cell, MHC Class I, MHC Class II and interferon-γ (IFN-γ) epitopes with high antigenicity scores. Moreover, membrane B-cell epitope (MBE) and envelope B-cell epitope (EBE) enhanced the overall stability, immunogenicity and antigenicity of the CoV-RMEN. The development of memory B-cells and T-cells was evident, with memory in B-cells lasting for several months. These finding opposed to several earlier reports where T-cell mediated immune response was considered a long-lasting response compared to B-cells (Abdelmageed et al., 2020; Wrapp et al., 2020). Another engrossing finding of this study was the development of Th1 response which enhances the growth and proliferation of B- cells augmenting the adaptive immunity (Carvalho et al., 2002). If a strong B-cell response occurred in animal trials (mice or rabbit), these antibodies could be used in diagnostic purposes, as they should recognize the prominent antigens on the viral surface (Kibria, Ullah & Miah, 2020). Moreover, CD8+ and CD4+ T-cell responses play major role in antiviral immunity (Abdelmageed et al., 2020). Another crucial fact is that Toll-Like Receptors (TLRs) can effectively bind with spike protein of the CoV (Totura et al., 2015; Zander et al., 2017), and might play an important role in the innate immune response to SARS-CoV-2 infection (Shahabi et al., 2020).

The physicochemical properties also revealed the chimera as a basic or alkaline protein (pI = 8.71) and would be thermostable upon expression, and thus, our proposed vaccine CoV-RMEN would be best suited for worldwide use in different endemic areas (Shey et al., 2019; Ul Qamar et al., 2019). The structural forms (secondary and tertiary) of the CoV-RMEN, when tested as the synthetic peptides, showed the ability to fold into their native structure, hence could mimic the natural infection by SARS-CoV-2 (Almofti et al., 2018). The refined tertiary (3D) structure of the final vaccine construct markedly presented the desirable structural features based on Ramachandran plot predictions (Shey et al., 2019; Srivastava et al., 2019). Molecular docking analysis showed that predicted chimeric protein can establish stable protein-protein interactions with TLRs (TLR-2, TLR-3, TLR-4) (Totura et al., 2015). An efficient activation of surface molecules of the CoV-RMEN is very crucial for immune activation of dendritic cells, and subsequent antigen processing and presentation to CD4+ and CD8+ T-cells via MHC-II and MHC-1, respectively (Shi et al., 2015; Shey et al., 2019; Shang et al., 2020). The molecular dynamics simulation also revealed that the docked CoV-RMEN-TLRs complexes were stable, and had more binding affinity TLR-3 and TLR-4 (Abraham et al., 2015). Furthermore, the CoV-RMEN showed good antigenicity scores on Vaxijen v2.0 and ANTIGENpro indicating that these peptide sequences are supposed to be highly antigenic in nature (Shey et al., 2019). The non-allergenic properties of the CoV-RMEN further strengthens its potential as a vaccine candidate (Shey et al., 2019; Ul Qamar et al., 2019).

Immune simulation of the CoV-RMEN exhibited expected results consistent with typical immune responses, and there was a growing immune responses after the recurrent antigen exposures (Fig. 6). The antiviral cytokine IFN-γ and cell stimulatory IL-2 level significantly increased, which also contribute to the subsequent immune response after vaccination in host (Almofti et al., 2018). This indicates high levels of helper T-cells and consequently efficient antibody production, supporting a humoral response (Shey et al., 2019). A lower IC50 value indicates higher binding affinity of the epitopes with the MHC class I and II molecules. While most of the previous studies (Sakib et al., 2014; Adhikari, Tayebi & Rahman, 2018) reported that a binding affinity (IC50) threshold of 250 nM identifies peptide binders recognized by T-cells, and this threshold can be used to select peptides, we kept binding affinity within 50 nM to get better confidence level in predicting epitopes for MHC-I and MHC-II alleles.The Simpson index estimated clonal specificity suggested a possible diverse immune response and this is plausible considering the generated chimeric peptide is composed of sufficient B- and T-cell epitopes (Fig. 6).

The interaction between T-cell epitopes, and their respective HLA alleles revealed significant binding affinity reflecting the immune activation of B- and T-cells as supported by other reports (Srivastava et al., 2019; Jaimes et al., 2020). T-cell epitopes from RBD and NTD regions showing high interaction with HLA alleles covered more than 98% of the world population with different ethnic groups, and these findings corroborated with many of earlier studies (Huang et al., 2007; Jaimes et al., 2020; Ul Qamar et al., 2019). The incorporation of GG and EGGE linkers between the predicted epitopes of the CoV-RMEN produced sequences with minimized junctional immunogenicity, and allowed the rational design construction of a potent multi-epitope vaccine (Huang et al., 2007; Badawi et al., 2016; Shey et al., 2019; Srivastava et al., 2019). The glycosylation of the surface antigens helps the enveloped viruses evade recognition by the host immune system, and can influence the ability of the host to raise an effective adaptive immune response (Pereira et al., 2018) or even be exploited by the virus to enhance infectivity (Wolfert & Boons, 2013). Moreover, some antibodies such as mAb 8ANC195 have evolved to recognize peptide epitope with no dependence on glycan binding (Kong et al., 2015). However, there is no data available for antibodies specific to spike glycoproteins of SARS-CoV-2, whether their recognition is interfered by the glycosylation of spike or may either be strengthened by sugars close to the peptide epitope, or not interfered by sugar modification (Zhou et al., 2020b). Furthermore, most of the epitopes of the CoV-RMEN harbor no glycoside apart from the NTD region (Watanabe et al., 2020). Furthermore, most of the glycans of the NTD epitopes present at the terminii of the putative HLA antigens, and may not interfere with the antigen presentation in an HLA complex (Grant et al., 2020). Integrity of the ACE2 receptor of the RBD, envelope protein B-cell epitope (EBE) and membrane protein B-cell epitope (MBE) in the CoV-RMEN suggests that the vaccine may maintain efficacy despite antigenic drift and glycosylation phenomena, as long as the virus continues to target the same host receptor (Grant et al., 2020).

One of the first steps in validating a candidate vaccine is to screen for immunoreactivity through serological analysis. This requires the expression of the recombinant protein in a suitable host. As we focused on the epitopes without the glycosylation or non-significant glycosylation, high-level expression of the vaccine was optimized into well-established and cost-effective prokaryotic expression system E. coli K-12 strain as the first choice using the plasmid pETite containing SUMO (Small Ubiquitin-like Modifier) tag and 6x-His tag facilitated both the solubilization and affinity purification of the recombinant protein (Biswal et al., 2015). Codon optimization of the CoV-RMEN revealed its high-level expression in E. coli (strain K12). Stable mRNA structure, codon adaptability index (0.87), and the GC content (50.26%) were favourable for high-level expression of the protein in the bacterium. After successful cloning of the gene, the recombinant plasmid can be propagated efficiently using E. coli cells, and subsequent protein expression can be performed in E. coli K-12 strain using IPTG (Isopropyl β-d-1-thiogalactopyranoside) induction, and cultivation at 28 °C as also reported earlier (Biswal et al., 2015).

As alternative to E. coli, eukaryotic cell line, HEK-293 was considered for CoV-RMEN expression. The codon adaption index (CAI), GC content were 1.0 and 61.60 respectively, wihch indicate high level of expression of vaccine construct in the HEK-293 cell line. In this case, pSec Tag2 mammalian expression vector could be used, which have secretion signal from the V-J2-C region of the mouse Ig kappa-chain for efficient secretion of the recombinant protein, C-terminal poly-histidine (6xHis) tag for rapid purification with C-termnal c-myc epitope for detection with an anti-myc antibody. To remove the affinity and detection tags (His tag and c-myc epitope) after purification, Factor Xa clevage site (LVPR↓GS) could be added to the C-terminal of the CoV-RMEN (Waugh, 2011).

Methods

Sequence retrieval and structural analysis

A total of 250 partial and complete genome sequences of SARS-CoV-2 were retrieved from NCBI (National Center for Biotechnology Information, https://www.ncbi.nlm.nih.gov/protein) (Table S5). We aligned these sequences through MAFFT online server (https://mafft.cbrc.jp/alignment/server/) using default parameters, and Wu-Kabat protein variability was analyzed (Fig. S9) in protein variability server (http://imed.med.ucm.es/PVS/) for SARS-CoV-2 NCBI reference genome (Accession no: NC_045512.2). We retrieved the S protein sequences of the SARS-CoV and MERS-CoV from the whole genome reference sequences of the respective three viruses from the NCBI database. Moreover, the S proteins of SARS-CoV (GenBank accession no: NC_004718.3), MERS-CoV (GenBank accession no: NC_019843.3) and SARS-CoV-2 were structurally compared using SWISS homology modeling (Waterhouse et al., 2018) based on the protein databank (PDB) templates 6acd, 5w9 h and 6vsb, respectively aligned using PyMOL (Faure et al., 2019), and observed for heterogeneous domains in the conformations. The models were optimized by energy minimization using the GROMOS96 program (Van Gunsteren et al., 1996) implemented within the Swiss-PdbViewer, version 4.1.0 (Guex & Peitsch, 1997). The Ramachandran plots of the derived models were evaluated using a PROCHECK (version 3.5)-server to check the stereochemical properties of the modeled structures (Laskowski et al., 1993). All further analyses including epitopes selection, antigenicity and allergenicity profiles, molecular docking and final vaccine construct were performed based on the NCBI reference genome of SARS-CoV-2.

Screening for B and T cell epitopes

Conformational B-cell epitopes on the S protein were predicted by ElliPro (http://tools.iedb.org/ellipro/) with the minimum score value set at 0.4 while the maximum distance selected as 6 Å(Kringelum et al., 2012). The linear B-cell epitopes of RBD and NTD regions of S protein, full length E and M proteins were predicted by “BepiPred Linear Epitope Prediction” (Larsen, Lund & Nielsen, 2006), and ABCPred with default parameters (Saha & Raghava, 2006). To find out the most probable peptide-based epitopes with better confidence level, antigenecity of the predicted peptides were further verified using VaxiJen antigenicity scores (Kringelum et al., 2012). The Kolaskar and Tongaonkar antigenicity profiling from IEDB analysis resource was also used for the RBD and NTD segments, E and M proteins (Kolaskar & Tongaonkar, 1990).

CTL epitopes, proteasomal cleavage and transporter associated with antigen processing (TAP), and HTL epitopes of the SARS-CoV-2 S, E and M proteins were predicted using IEDB resource tool Proteasomal cleavage/TAP transport/MHC class I combined predictor (http://tools.iedb.org/main/tcell/) with all default parameters . Moreover, the HTL epitopes of the proteins were screened using the IEDB tool “MHC-II Binding Predictions” (http://tools.iedb.org/mhcii/). Threshold for strong binding peptides (IC50) was set at 50 nM to determine the binding and interaction potentials of helper T-cell epitopes and both major histocompatibility complex (MHC) class I and alleles (Shi et al., 2015). Top five HLA epitopes for each RBD and NTD segments were docked against the respective HLA (MHC-I and MHC-II) allele binders by interaction similarity-based protein-peptide docking system GalaxyPepDock of the GalaxyWeb, docked HLA-epitope complexes were refined in GalaxyRefineComplex and binding affinity (ΔG) was determined PROtein binDIng enerGY prediction (PRODIGY) tool (Xue et al., 2016).

IFN-γ-inducing epitope prediction

Potential IFN-γ epitopes of all the selected antigenic sites of RBD, NTD, envelope protein B-cell epitope (EBE), and membrane protein B-cell epitope (MBE) were predicted by “IFNepitope” server (http://crdd.osdd.net/raghava/ifnepitope/scan.php). To identify the set of epitopes associated with MHC alleles that would maximize the population coverage, we adopted the “Motif and SVM hybrid” (MERCI: Motif-EmeRging and with Classes-Identification, and SVM) approach. The prediction is based on a dataset of IFN-γ-inducing and IFN-γ-noninducing MHC allele binders (Dhanda, Vir & Raghava, 2013).

Design and construction of multi-epitope vaccine candidate (CoV-RMEN)

The candidate vaccine (denoted as ‘CoV-RMEN’) design and construction method follows previously published peptide vaccine development protocol for different emerging infectious diseases like SARS and MERS (Shi et al., 2015; Badawi et al., 2016; Almofti et al., 2018; Shey et al., 2019; Srivastava et al., 2019; Ul Qamar et al., 2019). The multi-epitope protein was constructed by positioning the selected RBD, NTD, MBE and EBE aa sequences linked with short, rigid and flexible linkers GG. To develop highly immunogenic recombinant proteins, two universal T-cell epitopes were used, namely, a pan-human leukocyte antigen DR-binding peptide (PADRE) (Agadjanyan et al., 2005), and an invasin immunostimulatory sequence taken from Yersinia (Invasin) (Li et al., 2015) were used to the N and C terminal of the vaccine construct respectively, linked by EGGE.

Secondary and tertiary structure of the CoV-RMEN

Chou and Fasman secondary structure prediction server (CFSSP: https://www.biogem.org/tool/chou-fasman/), and RaptorX Property (http://raptorx.uchicago.edu/StructurePropertyPred/predict/) web-servers were used for secondary structure predictions (Källberg et al., 2014). The tertiary structure of the CoV-RMEN was built in homology/analogy recognition engine (Phyre2) (http://www.sbg.bio.ic.ac.uk/∼phyre2/html/page.cgi?id=index) web-server. The 3D model was refined in a three-step process, initially energy minimization using the GROMOS96 program implemented within the Swiss-PdbViewer (version 4.1.0). After energy minimization, the model was refined using ModRefiner (https://zhanglab.ccmb.med.umich.edu/ModRefiner/) and then GalaxyRefine server (http://galaxy.seoklab.org/cgi-bin/submit.cgi?type=REFINE).

Furthermore, the local structural quality of the CoV-RMEN was refined with GalaxyRefine server, and ProSA-web (https://prosa.services.came.sbg.ac.at/prosa.php) was used to calculate overall quality score for the refined structure. The ERRAT server (http://services.mbi.ucla.edu/ERRAT/) was also used to analyze non-bonded atom-atom interactions compared to reliable high-resolution crystallography structures. A Ramachandran plot was obtained through the RAMPAGE server (Lovell et al., 2003).

Physicochemical properties prediction of CoV-RMEN

Moreover, the online web-server ProtParam (Gasteiger et al., 2005) was used to assess various Physicochemical parameters of the CoV-RMEN including aa residue composition, molecular weight, theoretical pI, instability index, in vitro and in vivo half-life, aliphatic index, and grand average of hydropathicity (GRAVY). The solubility of the multi-epitope vaccine peptide was evaluated using the Protein-Sol server (https://protein-sol.manchester.ac.uk/).

Antigenicity, allergenicity and Immune simulation of the CoV-RMEN

VaxiJen 2.0 and ANTIGENpro (http://scratch.proteomics.ics.uci.edu/) web-servers were used to predict the antigenicity of the CoV-RMEN while the AllerTOP 2.0 (http://www.ddg-pharmfac.net/AllerTOP) and AllergenFP (http://ddg-pharmfac.net/AllergenFP/) web-servers were used to predict vaccine allergenicity. Further, the immunogenicity and immune response profile of the CoV-RMEN were characterized by in silico immune simulations using the C-ImmSim server (http://150.146.2.1/C-IMMSIM/index.php) under default parameters with time steps set at 1, 84, and 170 (each time step is 8 h and time step 1 is injection at time = 0). Therefore, three shots were given at four weeks apart.

Molecular docking and dynamics of the CoV-RMEN with TLRs

Molecular docking of the CoV-RMEN with the TLR2 (PDB ID:3D3I), TLR3 (PDB ID: 1ZIW) and TLR4 (PDB ID: 4G8A) receptors was performed using the High Ambiguity Driven DOCKing (HADDOCK, version 2.4) (De Vries, Van Dijk & Bonvin, 2010) web-server to evaluate the interaction between ligand (CoV-RMEN) and receptors (TLRs) leading to an enhanced immune response. CPORT (https://milou.science.uu.nl/services/CPORT/) was used to predict active interface residues between the CoV-RMEN and TLRs. For the analysis of the stable complex formation, all the complexes (TLRs with CoV-RMEN) were subjected to molecular dynamics (MD) simulation by Gromacs 2020.2 using OPLS-AA (Abraham et al., 2015; Jorgensen et al., 1996). Finally, the binding affinities of the docked chimeric protein-TLRs complexes were predicted using the PRODIGY (PROtein binDIng enerGY prediction) (https://nestor.science.uu.nl/prodigy/) web-server.

Analysis of cDNA and mRNA for cloning and expression of CoV-RMEN

Reverse translation and codon optimization were performed using the GenScript Rare Codon Analysis Tool (https://www.genscript.com/tools/rare-codon-analysis) in order to express the CoV-RMEN in the E. coli (strain K12). Stability of the mRNA was verified using two different tools, namely RNAfold (http://rna.tbi.univie.ac.at/cgi-bin/RNAWebSuite/RNAfold.cgi) and the mfold (http://unafold.rna.albany.edu/?q=mfold) web-servers. The optimized gene sequence of CoV-RMEN will be artificially synthesized having N-terminal recombinant human rhinovirus (HRV 3C) protease site (LEVLFQ↓GP) and cloned the final construct into pETite vector (Lucigen, USA) through enzyme-free method (Waugh, 2011). Finally, the sequence of the recombinant plasmid was designed by inserting the adapted codon sequences into pETite vector using SnapGene software (from Insightful Science; available at snapgene.com). As an alternative to E. coli, eukaryotic expression system HEK-293 was optimized using similar analysis for the vaccine production.

Population coverage by CTL and HTL epitopes

The predicted T-cell epitopes were shortlisted based on the aligned Artificial Neural Network (ANN) with half-maximal inhibitory concentration (annIC50) values up to 50 nM. The IEDB “Population Coverage” tool (http://tools.iedb.org/population/) was used to determine the world human population coverage by the shortlisted CTL and HTL epitopes (Bui et al., 2006). We used OmicsCircos to visualize the association between world population and different ethnic groups (Hoque et al., 2020b).

Conclusions

This multi-epitope peptide vaccine candidate, CoV-RMEN possesses potential epitopes from the RBD and NTD segments of spike (S), M and E proteins retaining potential antigenicity and non-allergenicity properties. This chimera, suitable for high-level expression and cloning, includes potential CTL, HTL and B-cell epitopes ensuring humoral and cell mediated immunity, as well as predicted immune-simulation refers to increased production of immunoglobulins and cytokines. Molecular docking and dynamic simulation of the CoV-RMEN with the immune receptors (TLRs) predicted strong binding affinity, in particular with TLR3 and TLR4. Remarkably, the CoV-RMEN had more than 90.0% world population coverage for different ethnic groups. The limitations posed by fewer number of SARS-CoV-2 geneome sequence data which tends to mutate frequently may not affect our analysis since we included four peptides of high conservancy from three major proteins of SARS-CoV-2 genome in a multi-epitope vaccine with the high conservancy. However, future in vitro and in vivo studies are required to assess the potentiality of the designed vaccine candidate to induce a positive immune response against SARS-CoV-2 infections, and also to validate the results obtained herein through immuno-informatics analyses.

Supplemental Information

Data S1 Data related to structural proteins of SARS-CoV-2 genomes

Click here for additional data file.

Data S2 Population coverage of geographic regions and ethnic groups

Click here for additional data file.

Table S1 NCBI partial or complete genome sequence accession numbers and their respective countries

Click here for additional data file.

Table S2 Predicted B-cell epitopes in RBD and NTD regions of S glycoprotein, envelop (E) and membrane (M) proteins of the SARS-CoV-2 through BepiPred-2.0 sequential B-Cell epitope predictor

The epitopes highlighted in green were considered promising vaccine candidates against B-cells of SARS-CoV-2.

Click here for additional data file.

Table S3 Predicted B-cell epitopes in RBD and NTD regions of S glycoprotein, envelop (EBE) and membrane (MBE) proteins of the SARS-CoV-2 through Kolaskar and Tongaonkar antigenicity profiling

Click here for additional data file.

Table S4 Predicted B-cell epitopes in RBD and NTD regions of S glycoprotein of the SARS-CoV-2 through ABCPred-2.0 B-Cell epitope predictor

Click here for additional data file.

Table S5 IFN- γ inducing epitopes predicted by IFNepitope program

Click here for additional data file.

Figure S1 Linear sequence alignment of the spike (S) proteins of the SARS-CoV-2, SARS-CoV and MERS-CoV

Using ClustalW multiple sequence alignment tool (version 1.2.4) we found that the S protein of SARS-CoV-2 shares 77.38% and 31.93% sequence identity with the S proteins of the SARS-CoV and MERS-CoV, respectively.

Click here for additional data file.

Figure S2 The Ramachandran plots derived from homology modeling

The three plots (a), (b) and (c) respectively illustrates the spike (S) proteins of SARS-CoV-2, SARS-CoV and MERS-CoV of Ramachandran outputs using PROCHECK web server. Most favored regions in the plots of are shown in red, additional allowed regions are shown in yellow, generously allowed regions are shown light brown, and disallowed regions are shown in white. After refinement, in SARS-CoV-2 S protein 85.1% and 12.7% amino acid residues were found in favored and allowed regions, respectively. The SARS-CoV S protein however had 78.4% and10.2% residues in favored and allowed regions, respectively, and 88.1% and 19.6% residues belonged to favored and allowed regions, respectively in MERS-CoV S protein.

Click here for additional data file.

Figure S3 The trimeric conformation spike (S) proteins

The S proteins of the SARS-CoV-2, SARS-CoV and MERS-CoV are trimeric conformation consisting of three homologous chains named as chain A, B and C . Structural alignment of these three chains using PyMOL revealed high degree of structural divergences in the N-terminal domains (NTDs) and receptor binding domains (RBDs) of the chains A and C compared to that of chain B.

Click here for additional data file.

Figure S4 Three-dimensional (3D) structure of the spike (S) proteins of SARS-CoV-2 (surface view)

The red, cyan, and yellow colored regions represent the potential antigenic domains predicted by the IEDB analysis resource ElliPro analysis whereas the gray colored region represents the transmembrane domain of S protein.

Click here for additional data file.

Figure S5 Predicted B-cell epitopes using Kolaskar and Tongaonkar antigenicity profiling in IEDB-analysis resource web-based repository

Yellow areas above threshold (red line) are proposed to be a part of B cell epitopes in (a) RBD and (b) NTD regions of S protein of the SARS-CoV-2.

Click here for additional data file.

Figure S6 Predicted secondary structure of CoV-RMEN using CFSSP:Chou and Fasman secondary structure prediction server

Click here for additional data file.

Figure S7 Graphical representation of solvent accessibility of CoV-RMEN vaccine candidate sequence

Click here for additional data file.

Figure S8 Graphical representation of the 3D structure validation of the CoV-RMEN vaccine candidate using ERRAT on-line server

Click here for additional data file.

Figure S9 Wu-Kabat protein variability plot for S, M and E protein

Variability plot of (a) spike (S) glycoprotein (b) membrane (M) protein and (c) envelope (E) protein.

Click here for additional data file.

The authors thank Joynob Akter Puspo, Masuda Akter and Israt Islam (MS student), and DR. Kazi Alamgir Hossain (PhD Fellow) of the Microbial Genetics and Bioinformatics Laboratory, Department of Microbiology, University of Dhaka for their cooperation, suggestions and overall encouragement during the preparation of the manuscript. The authors also extend their thanks to those who made their sequence data available in NCBI.

Additional Information and Declarations

Competing Interests

Author Contributions

Data Availability

Keith A. Crandall is an Academic Editor for PeerJ. The authors declare there are no competing interests.

M. Shaminur Rahman, M. Nazmul Hoque, M. Rafiul Islam, Salma Akter and ASM Rubayet-Ul-Alam performed the experiments, analyzed the data, prepared figures and/or tables, authored or reviewed drafts of the paper, and approved the final draft.

Mohammad Anwar Siddique analyzed the data, prepared figures and/or tables, authored or reviewed drafts of the paper, and approved the final draft.

Otun Saha analyzed the data, prepared figures and/or tables, authored or reviewed drafts of the paper, and approved the final draft.

Md. Mizanur Rahaman, Munawar Sultana, Keith A. Crandall and M. Anwar Hossain conceived and designed the experiments, authored or reviewed drafts of the paper, and approved the final draft.

The following information was supplied regarding data availability:

Accession numbers for the third-party sequences are available in Table S1.

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
