# Peer review of "Epitope-based chimeric peptide vaccine design against S, M and E proteins of SARS-CoV-2, the etiologic agent of COVID-19 pandemic: an in silico approach"

_PeerJ, doi:10.7717/peerj.9572_

## Round 0.1 · original submission · Major Revisions

Please consider all suggestions from the reviewers, including from those who recommended rejection of the manuscript and send us a manuscript with these modifications and a point-by-point rebuttal letter.

Reviewer 1 ·

Basic reporting

The article has lost its flow and presentation. It’s written so elaborately that following the analysis and results is difficult. Authors need to considerably shorten the article and make it to the point.

Experimental design

There are several flaws in methods, probably due to the description of the methods. It’s evident in each method they have described and their corresponding results. For example, It is unclear which virus spike structure was used to predict the epitopes and authors have taken SARS-CoV-2, SARS-CoV and MERS-CoV. For sequence base prediction also, how they have selected the specific sequence and why for epitope prediction is unclear. Is the sequence unique from SARS-CoV and MERS-CoV?? And so on.

Validity of the findings

Authors need reanalysis with proper biological objectives

Additional comments

Complete rewriting the article to make it short and understandable is required

Reviewer 2 ·

Basic reporting

The manuscript seemingly sounds timely or opportunistic given the crowd sourcing attached to current Covid-19 pandemic.

Several preprint servers are flooded with this kind of computational studies on the novel coronavirus proteins and epitope binding analyses. Further, a number of candidate vaccines have already entered in clinical trials. These efforts need to be covered in literature survey. And also the authors need to be precise as to how their efforts are uniquely posited in comparison to the other efforts such that they be published in Peer J.

The language and presentation is poor and the manuscript could have been copy edited by a professional editing agency prior to its submission.

Experimental design

The design is of simplistic homology modeling, docking and simulation.

This reviewer would have liked a molecular dynamic simulation at femtosecond resolution scale.

Experimental approaches at high level expression and recombinant protein production are necessary to prove if the modeled molecules are indeed expressable or biologically active or stable.

Validity of the findings

Computational models could be misleading unless they are validated through molecular dynamic simulation and lab based verification.

Additional comments

None

·

Basic reporting

The article is well structured and written in clear, unambiguous and technically correct English. It is easy and pleasant to read.

Literature references, sufficient field background/context provided.

The article perfectly contextualizes the research and fully justifies it, using very appropriate and up-to-date literature.

Professional article structure, Figures, Tables. Raw data shared.

Regarding Figures, both, those in the main text and those included in the Supplementary material, are adequate and help the understanding of the text. They are well numbered and correlate with the text. However, they should be of a higher quality due to the amount of material included in each Figure. In this way, Figures such as 4, 5 or 7 may be difficult to see if the quality is not improved. In any case, a higher quality of all the Figures would be necessary.

1. In Figure 1, line 2, change "MERS-CoV is a trimeric conformation" by "MERS-CoV are trimers".
2. In Figure 1, line 3, delete the space after "C".
3. In Figure 1, line 3, change "Respective sequences of these three chains were aligned and visualized using Pymol" by "Structural alignment of these three chains using Pymol".
4. In Figure 1, line 5, change "of the chain A and C" by "of the chains A and C".
5. In Figure 2, line 4, change "Elipro" by "Ellipro".
6. In Figure 3, line 4, delete "While green areas are not".
7. Figure 4 needs some more discussion in the main text about docking and the presence of glycosylations. Additionally, higher quality is required in order to see all the details in the Figure.
8. In Figure 5, higher quality is required.
9. In Figure 6, line 4, delete the space after "server".
10. In Figure 7, panel f, change placement of the legend.
11. In Figure 8, panel d, align this panel with the rest of the panels (e and f).
12. In Supplementary Figure 8, change “Envelop” by “envelope”.

Regarding the Tables, they are also suitable for the understanding of the text, however, the numbering of the Supplementary tables does not correspond to the numbering in the main text.

Regarding Tables and Figures, it would be helpful if the numbering of the S protein sequences is maintained throughout the text, Figures and Tables. For example, while in Table 1, the protein S sequence is the normative one, in the rest of the Tables, Figures and text, it is numbered differently (considering as amino acid 1 the beginning of both NTD and RBD), which leads to confusion when locating the different epitopes in the full length sequence of S protein. In this way, is recommended to clearly indicate this issue somewhere in the text to avoid confusion. Also as an alternative I would propose a Figure showing the full sequence of the S protein showing both the actual numbering and the numbering used by the authors.

Self-contained with relevant results to hypotheses

The article is self-contained, and is organized as a publishing unit, with its respective hypothesis and corroborating results. There are a lot of good looking and organized results that together form an interesting article.

Experimental design

Original primary research within Aims and Scope of the journal.

The article by Rahman et al., meets the Aims and Scope of the journal and I truly think it could be suitable for publication in PeerJ.

Research question well defined, relevant & meaningful. It is stated how research fills an identified knowledge gap.

The article pursues an ambitious objective which is very relevant in the current health emergency situation by COVID-19. The development of a vaccine against the SARS-CoV-2 virus is of vital importance and I therefore believe that the article can give some clues on how to tackle this issue. The issue is therefore clearly defined, as is the objective of the paper and, as I mentioned above, it is extremely relevant due to the lack of information on the subject.

Rigorous investigation performed to a high technical & ethical standard.

The research has been rigorously performed following high technical standards although I have some concerns that are exposed as Major points in this document.

Methods described with sufficient detail & information to replicate.

The methodology set out in the article is clear, comprehensive and well explained, which suggests to me that the results of the article would be replicable.

Validity of the findings

Impact and novelty not assessed. Negative/inconclusive results accepted. Meaningful replication encouraged where rationale & benefit to literature is clearly stated.

The impact of the article is evident in the current situation and therefore the research is clearly relevant. As for the novelty, the use of multi-epitope approaches in vaccine design is quite novel and the literature on this topic has been growing significantly in the recent years. In fact, as discussed in the article, the presence of several epitopes in a single peptide avoids the need to use attenuated viruses and in turn allows the generation of an effective immune response against various viral protein epitopes. The results obtained are promising although I have some concerns regarding the role that glycosylations present in the viral protein and that are not considered in the design and elaboration of the experimental research, may have on the final result (See major points).

All underlying data have been provided; they are robust, statistically sound, & controlled.

All the data are provided both in the main Figures and Tables and in Supplementary material.

Conclusions are well stated, linked to original research question & limited to supporting results.

Conclusions clearly match the obtained results and answer the original research question initially proposed. My only doubt refers to the absence of glycosylations of the S protein throughout the paper and the role these may have in shielding the viral S protein from the antibodies generated as a result of the immunization with the proposed CoV-RMEN peptide. Additionally, if the glycosylated version is required, then, vaccine production will be much more expensive and not relatively cheap as exposed by the authors in the final conclusions.

Additional comments

Major and minor points to be addressed

The article by Rahman et al., is of great interest and in fact the proposed approach is attractive and feasible at a time. The article is well written, clear, and despite the large number of results, it is well organized. In general I think the approach is correct, but I have some comments that may be relevant and need author’s attention.

Major points

1. The main point against this work is the lack of evidences about the existing glycosylations in the Spike protein as a parameter that can affect the antigenicity of the proposed construction. In fact, this has considerable implications throughout the article. Thus, for example, in Figure 4 where the interaction of various NTD epitopes is modelled. 7 of the 10 epitopes shown in this Figure have some glycosylation (which have been previously published in Site-specific glycan analysis of the SARS-CoV-2 spike. Watanabe et al., 2020), which may affect their recognition, since glycosylations may hide certain epitopes. In fact, the proposed final construction (CoV-RMEN) has 139 amino acids from the NTD, which contains 5 asparagines that are glycosylated in the viral protein (Watanabe et al., 2020). In fact, some of the residues proposed as relevant in the interaction of CoV-RMEN with TLR2-4 are actually close to glycosylated residues in the viral protein and therefore, it would not be unreasonable to think that such epitopes, which would be functional in the proposed construct, would not be accessible in the viral protein. Examples of this are N122 (according to the complete sequence of the S protein of the virus), which is glycosylated (Watanabe et al, 2020), is located between the amino acids L101, I103 and I112, which are relevant in the interaction of CoV-RMEN with TLR3. Another example would be N165 (according to complete sequence of the virus S protein), which is glycosylated (Watanabe et al., 2020) and is the previous amino acid to C150 in the CoV-RMEN construct, which interacts with TLR3.

2. The effect of glycosylations also has a side effect in the selected construct expression system. If what we are interested in is a construct as similar as possible to the real viral protein, we could consider the 5 glycosylations present in the proposed construct as important. If so, the expression system could no longer be E.coli, since it is not capable of reproducing such post-translational modifications. In this case, a HEK-type expression system, for example, would be more appropriate. I think these observations should be added and discussed in the article.

3. In the 3D model proposed for CoV-RMEN, neither the SUMO tag nor the histidine tag are included, right? How would the presence of both tags affect the performed dockings? has the possibility of adding a site for a protease (for example 3C) considered for a cleaner and more efficient purification and thus avoid the possible effect of the tags?.


Minor points

1. Lines 7, 11 and 15. The dash between the city and the postal code is missing.
2. Line 61. Change “April 15” to “April 15th”.
3. Lines 71 and 82. “Genus” and “Scientists” in lowecase.
4. Line 101. Change “stimulates” by “stimulating”.
5. Lines 122-124. Change the sentence "Multi-epitope vaccine candidates have already been designed..... and their efficacies have been further reported in MERS-CoV and SARS-CoV". In fact, the cited reports are just bioinformatic approaches like the one we are dealing with here and I sincerely do not think that the authors can say that these types of vaccines are effective with the references they provide.
6. Lines 138, 200, 207, 446, 470, 572, 648 and legends for Figures 3, 6, Supplementary Table 2, 3 and Supplementary Figure 8. Change “envelop” by “envelope”.
7. Line 141. It is not clear to me why the 3D conformation of S protein is predicted, when S protein 3D structure itself is used as a model, with the same starting sequence as that used by the authors. If the model has been made to complete the regions not modelled in the 6vsb structure, this should be indicated at some point, otherwise it is difficult to understand.
8. Lines 144-145. Remove one line.
9. Lines 155-157. Is the high degree of structural divergence referred to the SARS-CoV2, SARS-CoV and MERS-CoV S proteins? If so, it should be explained a little better and include the reference (Walls et al., 2020, Cell) both in Results and Discussion sections, explaining why this divergence is so high. Another option is to remove any reference to SARS-CoV or MERS-CoV as they do not ultimately contribute to anything essential in the article.
10. Line 162. Add Fig.1 after “MERS-CoV” to clarify the sentence.
11. Lines 169-171. Would it be possible to re-order the epitopes according to their position in the sequence and not according to their score? I say this because it is more intuitive and easier to compare the same epitopes among proteins.
12. Line 174. Change “Elipro” to “Ellipro”.
13. Line 180. Add "prediction" after “B-cell epitopes”.
14. Lines 188, 205, 208 and 252. Supplementary Tables 1, 2, 3 and 4 are not properly labelled.
15. Line 224. Add a space between "13-mer" and "124".
16. Line 238. Define "aa".
17. Line 246. Add "membrane protein" before "B-cell linear epitope".
18. Line 309. Remove "(0.0%)".
19. Lines 327-330. Remove 3 lines.
20. Lines 353 and 353. Remove 1 line.
21. Lines 354-367. In Table 3, TLR2 is also shown. Nevertheless, not a single mention is done along the paragraph.
22. Line 399. Remove one of the parentheses from "((Biswal et al., 2015).
23. Lines 401 and 402. Remove 1 line.
24. Discussion section should be shorter. We can find several parts of the Discussion section where results are repeated and could be removed (for example lines 474-480). It would be interesting to make the discussion a little more concise and in any case include some of the comparison between SARS-CoV-2, SARS-CoV and MERS-CoV to ultimately meet the interest of comparing the physicochemical parameters of the S proteins of the three virus species (point 8). In addition, in lines 494-496 the different antigenicity of CoV-RMEN is mentioned depending on the program used but nothing is discussed about this fact.
25. Line 519. Change "enhance" by "enhances".
26. Line 549-553. Delete 4 lines.
27. Line 571. "involved in".
28. Line 660. " Glutamate should be in lowecase.
29. Add one space line between lines 665 and 666.
30. Lines 745-762. Take care about the quantity of empty lines between the two sections.
31. Lines 773-774. Delete 1 line.
32. Reference section. Please check the reference style as there are some errors (for example, some volume numers are in bold and some other are not).

---

## Round 0.2 · Minor Revisions

The manuscript is much improved, please correct these new suggestions.

·

Basic reporting

The authors have answered efficiently to all my queries, generating a sufficiently improved version to be considered for publication in PeerJ.

However, there are still some minor points that need to be addressed before publication:

Minor points:

- Line 179: remove the hyphen between amino-acids
- Line 182: In vivo should be in italics
- Line 292: Remove a space between "et al., 2020). and Although".
- Lines 346, 353, 356, 407, 431 and line 3 in figure 5 footnote: Change CovRMEN to CoV-RMEN

Experimental design

No comment

Validity of the findings

No comment

Additional comments

No comment

---

## Round 0.3 · accepted · Accept

I believe that now it only needs to be technically evaluated by our office and the manuscript is accepted.